# Consumers’ Neophobic and Variety-Seeking Tendency in Food Choices According to Their Fashion Involvement Status: An Exploratory Study of Korean Consumers

**DOI:** 10.3390/foods12091878

**Published:** 2023-05-02

**Authors:** Sangwoo Seo, Mina K. Kim

**Affiliations:** 1Department of Fashion Business, Jeonju University, 303 Cheonjam-ro, Wansan-gu, Jeonju-si 55069, Republic of Korea; sangwooseo@jj.ac.kr; 2Department of Food Science and Human Nutrition, Jeonbuk National University, 567 Baekjedaero, Deokjin-gu, Jeonju-si 54896, Republic of Korea

**Keywords:** food neophobia, variety-seeking scale, fashion involvement, fashion

## Abstract

With recent food innovation and technological advances, a considerable number of new food products are being developed and launched in the global food market, and various attempts have been made to collaborate between food and fashion businesses to achieve a competitive edge. Fashion and food are essential items in our daily lives, so people are intentionally and unintentionally exposed to consumption decisions regarding these two items on a regular basis. The objective of this study was to determine consumers’ neophobic and variety-seeking tendencies in food choices according to their involvement in fashion-related choices. Internet surveys were conducted (*n* = 215), which included questionnaires regarding the food neophobia scale (FNS), the variety-seeking scale (VARSEEK), and the fashion involvement scale (FIS), along with demographic-related questions. A negative correlation was observed between the FNS and FIS (r = −0.735, *p* < 0.0001), suggesting that consumers who are highly involved in fashion product choices have neophobic tendencies. A positive correlation was observed between VARSEEK and FIS scores (r = 0.353, *p* < 0.0001), as was expected from FNS and FIS results. No significant differences in demographic characteristics between those with high and low FIS scores were observed, suggesting that other factors may have influenced these results.

## 1. Introduction

In light of recent food innovation and technological advances, a considerable number of new food products are being developed and launched in the global food market [1]. The product life cycle (PLC) is becoming increasingly shorter for newly launched food products. Similarly, fast fashion has become a keyword in the fashion industry [2], with shorter PLCs observed for both inexpensive and high-end apparel [3]. In the competitive market, various collaborative attempts have been made between food and fashion businesses to achieve a competitive edge. For example, the global fashion brand Nike and ice cream manufacturer Ben & Jerry’s collaborated in 2020 to create the “Chunky Dunky” limited edition in the Nike SB Dunk Low shoe line, inspired by the colorful design of Ben & Jerry’s ice cream packaging and logo. As another example, FILA Korea collaborated with Pepsi to launch various fashion items, including apparel, backpacks, and shoes, to create a young and vibrant brand image. In addition to these two instances, many collaborations have been made between food and fashion companies in recent years, and such collaborations have created new markets and attracted new customers to each category.

Involvement refers to a general measure of orientation that captures consumer views on the central part of their lives [4,5,6]. Fashion involvement refers to the extent of interest in time, money, and attention spent on fashion product categories [7]. Therefore, it can, directly and indirectly, measure people’s consumption behaviors toward fashion-related items, such as apparel [7,8]. While fashion involvement measures one-dimensional consumption attitude at the time of purchase, food involvement measures are much more complex due to the multidimensionality of the food consumption process, which includes acquisition, preparation, cooking, eating, and disposal [5,6]. Based on the definition of involvement, measuring consumer involvement in certain items (i.e., fashion apparel and foods) seems not only to measure individual interests in a particular item but also one’s value in consumption behaviors, which include demographic status, personal traits, consumption contexts, individual circumstances, and social responsibilities [9,10,11].

Due to the complexity of food involvement behaviors, different measures of reflecting food choice behaviors, such as the food neophobia scale (FNS) [12] and the variety-seeking scale (VARSEEK), are utilized. Food neophobia is an inherent adaptive personal trait of reluctance to eat and/or the avoidance of new foods [13,14]. Factors influencing food neophobia have been investigated from various perspectives, including age, gender, social status, urbanization, and cultural and social influences. The individual personality is also another factor influencing food neophobia, such that individuals with sensation-seeking tendencies show low food neophobia [15,16,17,18]. The variety-seeking (VARSEEK) tendency can be viewed as the opposite of food neophobia, as individuals with high VARSEEK tendencies are more actively engaged in and motivated to seek variety in their food choices [5]. Studies have reported a high negative correlation between the FNS and the food involvement scale, indicating that measuring consumer food neophobic tendencies can be another way of understanding consumer behaviors in food choices. Individual involvement in food and/or fashion can measure the consumer values associated with purchase and consumption decisions, which are often influenced by individual traits.

While fashion involvement and food neophobia are not directly related, they both reflect the way consumers approach different aspects of their lives. To date, food-related research among people in the fashion industry and/or “fashion geeks” has scarcely been investigated. Among those limited studies, the majority were conducted on eating disorders among fashion models [19,20]. Besides professional fashion models, more people are aware of fashion and style trends, as people are exposed to digital media, where fashion-related information is abundant. Food and fashion are essential items in our daily lives, so people are intentionally and unintentionally exposed to consumption decisions regarding these two items on a daily basis. Nevertheless, studies connecting consumer behaviors in fashion and food choices are limited.

We hypothesized that people who are seeking high fashion trends may show variety-seeking tendencies in their food choices and will thus be less food neophobic. In other words, personal interest and a sensation-seeking personality in fashion may influence food neophobia and other food choice behaviors. To date, food choice behavior, including neophobic tendencies among people with different degrees of fashion involvement, has not been investigated. Understanding food choice behavior based on consumers’ fashion involvement status may provide valuable information to food marketers and product developers to specifically design food products that appeal to individuals who are open to trying new and different foods and fashion items. Therefore, the objective of this study was to determine consumers’ neophobic and variety-seeking tendencies in food choices according to their involvement in fashion-related choices.

## 2. Materials and Methods

This survey was conducted using a third-party market research agency to recruit a wide range of participants. The survey was uploaded to the Internet and launched on an online consumer database maintained by the market research agency. Prior to asking questions on food neophobia and fashion involvement, participants were first asked to answer a demographic background questionnaire that required them to provide their age, gender, height and weight, food consumption habits, and fashion-related consumption habits. Then, questionnaires regarding food choice, such as the FNS and VARSEEK, were included to capture individuals’ food choice behaviors. These questions were modified from a previous study conducted by a previous study [5]. Ten FNS questions and eight VARSEEK questions were asked. The FNS questions were measured using a 7-point category scale, with 1 defined as completely disagree, 4 defined as neither agree nor disagree, and 7 defined as completely agree. The VARSEEK measurement was conducted using a 5-point category scale, with 1 defined as completely disagree, 3 defined as neither agree nor disagree, and 5 defined as completely agree. A series of fashion involvement questionnaires [21] were included and were rated on a 7-point scale, with 1 defined as strongly disagree and 7 defined as strongly agree. As an incentive for participation, a draw for three USD 50 gift cards was held after all the participants had gone through the survey.

Frequency analysis was conducted on the collected data regarding demographic and consumption characteristics using XLSTAT (version 2020, Addinsoft, Paris, France). The FNS score was calculated using the method suggested by a previous study [5], where five out of the ten questions were reversely calculated. Fashion involvement scores (FISs) were calculated by averaging the mean values of six questions. An analysis of variance (ANOVA) was conducted on the FNS, VARSEEK, and FIS questions, followed by Fisher’s least significant difference (LSD) post hoc test (*p* < 0.05). The correlation coefficient was calculated between the FNS and VARSEEK, FNS and FIS, and VARSEEK and FIS, and the significant difference was determined by the calculated *p*-value. Ethical approval was also obtained.

## 3. Results

### 3.1. Participant Demographics and Consumption Characteristics

A total of 215 consumers voluntarily participated in the Internet survey; details of the participants’ demographics can be found in Table 1. Gender and age were almost equally distributed: 49.3% male, 50.7% female, and 15–16% in each age group. Participants in this survey had a variety of occupations, with monthly incomes fairly equally distributed and ranging from about USD 1000 to over USD 5000 per month. Despite the efforts to recruit participants from different regions in Korea, 62.8% of the participants were recruited from Seoul/Gyeonggi Province, followed by 21.4% from Gyeongsang (southeast), 7.0% from Chungcheong (central), 6.5% from Jeolla Province (southwest), and 2.3% from Gangwon Province (northeast). The mean value of the participants’ body mass index (BMI) was 23.2, and the values ranged from 11.0 to 38.2.

Table 2 shows the participants’ consumption characteristics of food and fashion-related items. Questionnaires were purposely designed to compare the consumption characteristics of food and fashion-related items, and the results of this comparison are presented in Table 2. In this section, the consumption characteristics of food specifically indicate “eating-out (dining-out)” occasions. For purchase frequency, slight differences between food and fashion-related items were observed. About 63.2% of the participants responded that they eat out more than three times a month, while the majority of the participants (85.6%) purchased fashion-related items less than two times a month. Time of purchase was made in dinner (73.0%), and lunch (20.5%) for eating out. Similarly, the majority of fashion-related item purchases were made around evening time (4–8 pm; 48.8%), followed by afternoon (12–4 pm, 31.6%), and late night (8 pm–12 am; 14.9%). The distribution of monthly spending budget for food and fashion-related items were similar, in that 34.0% (food) and 42.8% (fashion) responded that they spend <USD 100 per month; 37.7% (food) and 37.2% (fashion) spent USD 100–200 per month; 17.2% (food) and 12.1% (fashion) spent USD 200–300 per month. Flavor (71.2%) and design (49.8%) were the most important factors affecting their choice of dining place and fashion-related items, followed by price (18.1% for food and 27.0% for fashion-related items), respectively. For fashion-related items, 12.1% of the participants responded that convenience is also an important factor affecting their choice.

### 3.2. FNS and VARSEEK Results

Prior to intensive analysis, Cronbach’s alpha value was determined to ensure the reliability of the FNS and VARSEEK questionnaires. Cronbach’s α was 0.870 for the FNS, and 0.807 for the VARSEEK questionnaire. Therefore, all the questions under the FNS and VARSEEK were reliable in this survey. The mean values for each measure, as well as the calculated values of the FNS and VARSEEK, can be found in Table 3 and Table 4.

The FNS score was calculated by adding the average score for each statement belonging to the FNS. The FNS score was 30.61 in this study and was compared with the FNS scores from other countries: 38.5 for Indonesia, 37.4 for the Philippines, 39.3 for Malaysia, 34.1 for Vietnam, 26.1 for the UK, and 34.7 for Australia [14]. The FNS scores from this study on Korean participants were relatively lower than the FNS scores of consumers in other Asian countries, which suggests that Korean consumers tend to be more open to accepting new foods than consumers in other Asian countries.

While data are not presented in this manuscript, there were no substantial differences in the FNS based on participants’ age or gender (*p* > 0.05). The influence of age on individual FNS scores remains a topic of debate, as some studies suggest that there are no significant differences, while others suggest otherwise. According to the existing literature, research has found that neophobic tendency starts from weaning ag [22], increases until adolescence (age 13) [22,23,24], and then decreases from adolescence to adulthood [25]. Furthermore, elderly individuals (ages 66 and older) display slight increases in neophobia, which may be related to their health concerns, suggesting that stronger health status may decrease neophobic behavior [24]. Similarly, gender differences in food neophobia have been reported with varying conclusions in the literature. While one study found that men are more neophobic than women [25], another reported the opposite [26]. While the relationship between age and gender and food neophobia is still controversial, the results from this study indicate no significant age or gender effects on FNS scores (*p* > 0.05). In addition, FNS scores were compared based on the participants’ current residency (rural versus urban), and no significant differences were observed, which contradicts a prior study conducted on Australian adolescents [27].

In this study, the VARSEEK score, which was calculated by summing up the mean values of each questionnaire in the VARSEEK questionnaire, was 25.49. It has been widely acknowledged that food neophobia and variety-seeking exhibit diametrically opposed tendencies: While neophobia is characterized by the rejection of food variety, variety-seeking involves the acceptance of food variety [5,28]. When comparing the values collected from this study with UK participants, the current study on Korean consumers demonstrated lower VARSEEK scores, with UK participants recording a VARSEEK score of 29.33. This finding is consistent with FNS results, as the lower level of food neophobia among UK participants likely contributed to the higher VARSEEK score than the present study. VARSEEK is another metric for assessing openness and sensation-seeking tendencies, and individuals with high VARSEEK scores are typically less prone to neophobia, which is defined as an aversion or reluctance to try novel and unfamiliar foods [12]. 

### 3.3. FIS Results

For fashion involvement scores, Cronbach’s α value was 0.970, meaning that all the questions under the fashion product involvement category were reliable, and the results are listed in Table 5. The fashion involvement scale contains questionnaires that gauge the extent to which consumers are invested in fashion products, particularly clothing [4]. In a previous study, it was reported that product involvement had a strong correlation with consumption involvement (r = 0.98), purchase decision involvement (r = 0.89), and advertising involvement (r = 0.80), suggesting that this can be another way of measuring the dynamics of consumer behavior. Among the various consumption involvement scales, this study utilized a fashion involvement scale to indirectly assess the participants’ consumption characteristics. The mean value of the FIS scores was 3.81 on a seven-point scale. While a previous study suggested that more females than males and younger than older consumers tend to exhibit greater involvement in the fashion product consumption process [4], this study did not find any significant differences in gender or age group (*p* > 0.05, data not shown). Gaining a comprehensive understanding of consumer involvement is crucial, as research has demonstrated that highly involved consumers tend to exert more effort to customize their product selection by more actively communicating their preferences and dislikes about products [29].

### 3.4. Correlations between FNS, VARSEEK, and FIS

Table 6 presents the correlation matrix among the FNS, VARSEEK, and FIS. The results indicate a significant negative between the FNS and VARSEEK (*p* < 0.0001), which is in line with a previous study [5]. Specifically, the correlation coefficient of −0.735 (*p* < 0.0001) indicates that consumers with higher neophobic tendencies are less likely to seek variety in food choices. Additionally, the negative correlation between the FNS and FIS suggests that consumers with a greater degree of fashion involvement have lower levels of food neophobia (r = −0.178; *p* = 0.009). This finding implies that individuals with more active fashion product consumption behaviors may also be more willing to experiment with novel food products. The positive correlation between VARSEEK and FIS scores further supports this notion (r = 0.353, *p* < 0.0001), as individuals with high openness to sensation-seeking tendencies are more likely to be involved in both fashion and food choices. These results are consistent with a previous study conducted on UK college students, which found negative and positive correlations between the food involvement scale and FNS (−0.273; *p* < 0.01) and VARSEEK (r = 0.591; *p* < 0.01), respectively [5]. Notably, their study employed the food involvement scale, whereas our study utilized the fashion involvement scale to indirectly measure participants’ consumption characteristics. 

### 3.5. FNS and VARSEEK Tendencies According to FIS Status

In order to analyze the relationships between consumer behavior according to FNS and VARSEEK results regarding the levels of fashion involvement status, the participants were classified into three distinct groups based on their FIS scores: low, medium, and high. The classification criteria were established according to FIS scores, with the low category including participants with mean scores of 2.6 or less, the medium category comprising those with a score of 2.7 to 4.8, and the high category including those with a score of 4.9 or more. Each of these categories represented different quartile values, with the low group representing the lowest 25% (Q1), the medium group representing 25–75% (Q2), and the high group representing the highest 75% (Q3) FIS scores. The results of the comparison of FNS and VARSEEK scores across different fashion involvement categories are displayed in Figure 1a (FNS) and Figure 1b (VARSEEK). Remarkably, consumers with low FIS scores exhibited a higher food neophobic tendency, with an FNS score of 33.64, which was significantly higher than consumers in other groups (medium = 30.14, high = 28.25; *p* < 0.05). This indicates that individuals who demonstrate little interest in fashion product purchase decisions may also exhibit a greater tendency toward food neophobia. Similarly, consumers with low FIS scores displayed lower levels of variety-seeking tendencies in their food choices (23.24) than those in other groups (medium = 25.36, high = 28.21; *p* < 0.05).

Upon analyzing the demographical distribution of the individuals categorized into low, medium, and high groups, slight variances were detected (data not displayed). Notably, the individuals in the high group who exhibited high VARSEEK and low FNS scores displayed distinct demographic and consumption characteristics, differing from the other consumer groups. Specifically, 41.5% of these individuals were aged between 40 and 59 years, 43.4% had a monthly income exceeding USD 3000, and most significantly, a significantly higher proportion of this group (39.6%) prioritized price, with name value/brand (18.9%) being the second most important factor in influencing their food choices. In contrast, the other groups (medium and low) exhibited a higher preference for flavor (70.2% for the medium group and 63.8% for the low group) and price (19.2% for the medium group and 24.1% for the low group). Additionally, a higher proportion of individuals in the high group were found to dine out near downtown areas (17%) when compared with the medium (11.5%) and low (5.2%) groups. It is worth noting that the demographic and consumption characteristics of individuals in the medium and low groups displayed similar consumption patterns, differing from the high group. In particular, price (39.6%), flavor (26.4%), and name value/brand (18.9%) were important for the high group, whereas flavor (70.2% for the medium group and 63.8% for the low group) and price (19.2% for the medium group and 24.1% for the low group) were the top two factors influencing the food choices of individuals in the medium and low groups. Furthermore, 26% of the individuals in the medium group reported eating out during lunchtime, with a higher proportion of the individuals in all groups reporting eating out near their residence/school/workplace (approximately 63.5%). It is noteworthy that these proportions were significantly higher than the responses from the other groups.

## 4. Discussion

In the field of fashion involvement, researchers typically rely on four different measures: product involvement, advertising involvement, purchase involvement, and purchase decision involvement [4]. While these measures tend to be interrelated, the present study specifically focused on fashion product involvement as a representative measure of fashion involvement. Fashion product involvement refers to an individual’s interest and engagement in activities related to fashion, such as browsing, purchasing, and wearing clothing and accessories. Individuals who exhibit higher levels of fashion product involvement tend to be more interested in exploring a broad range of lifestyle choices, including those related to fashion. The results of this study support this hypothesis, with a negative correlation between FNS and FIS and a positive correlation between VARSEEK and FIS, thus providing further evidence for the validity of the hypothesis.

There are various factors that can contribute to neophobia, including demographic characteristics such as age and gender as well as personal traits and life stage. This study sheds light on another potential factor that may influence neophobic tendencies: an individual’s level of fashion involvement. Specifically, the findings indicate that consumers who exhibit high levels of fashion involvement tend to show lower levels of neophobia. While previous research has suggested that gender and age play a significant role in determining one’s level of fashion involvement [4,30,31], this study reveals that this is not always the case. In fact, a considerable proportion of the participants classified as highly involved in fashion products were aged between 40 and 59, suggesting that age may not be a universal determinant of fashion involvement tendencies.

Other studies have shown that personal traits, such as self-esteem and self-concept, can also influence an individual’s level of fashion involvement. For example, research has suggested that fashion choices, particularly apparel, can serve as a reflection of an individual’s personality and self-esteem in society. Those with low self-esteem may tend to conform to societal norms, while those with high self-esteem may use fashion as a means of expressing their unique personalities [32]. Additionally, previous research has indicated that neophobic tendencies in children are associated with lower levels of social, physical, and academic self-concept/esteem [33]. Taken together, these findings suggest that individuals with lower levels of fashion involvement may be more likely to exhibit neophobic tendencies due to lower levels of self-esteem.

In addition to the aforementioned factors, it is pertinent to acknowledge that cultural and societal elements may also have a substantial impact on the association between fashion involvement and food neophobia. Certain cultures may give significant weight to traditional food choices, which could potentially modify the relationship between fashion involvement and food neophobia. Studies in fact reported that education level may influence food neophobic tendencies, and social facilitation from parents, peers, and significant others eventually influences food choice behaviors [25,34,35], suggesting that societal influences can affect personality traits. Likewise, societal pressure to conform to specific beauty standards may also influence individual attitudes toward both food and fashion. Further research is imperative to scrutinize this relationship in greater depth by utilizing a rigorous survey design and to examine the role of culture and societal norms in the association between fashion involvement and food neophobia.

## 5. Conclusions

This study aimed to explore the relationship between fashion involvement and consumers’ neophobic and variety-seeking tendencies in food choices. The survey method was used to capture the respondents’ fashion involvement status and their food choice preferences. The findings of this study suggest that individuals with high fashion involvement exhibit lower levels of food neophobia and greater variety-seeking tendencies in their food choices. This study is the first attempt to establish a link between fashion involvement and food neophobic tendencies, and as such, several aspects require further investigation. For instance, the study does not offer a definitive explanation of how socioeconomic characteristics of individuals with high, medium, and low fashion involvement status correspond to low, medium, and high neophobic tendencies. This suggests that other factors, such as personality traits, self-esteem, social status, and cultural and societal elements may also influence consumers’ fashion and food choices. Therefore, a carefully designed survey questionnaire is required to explore these factors further.

## Figures and Tables

**Figure 1 foods-12-01878-f001:**
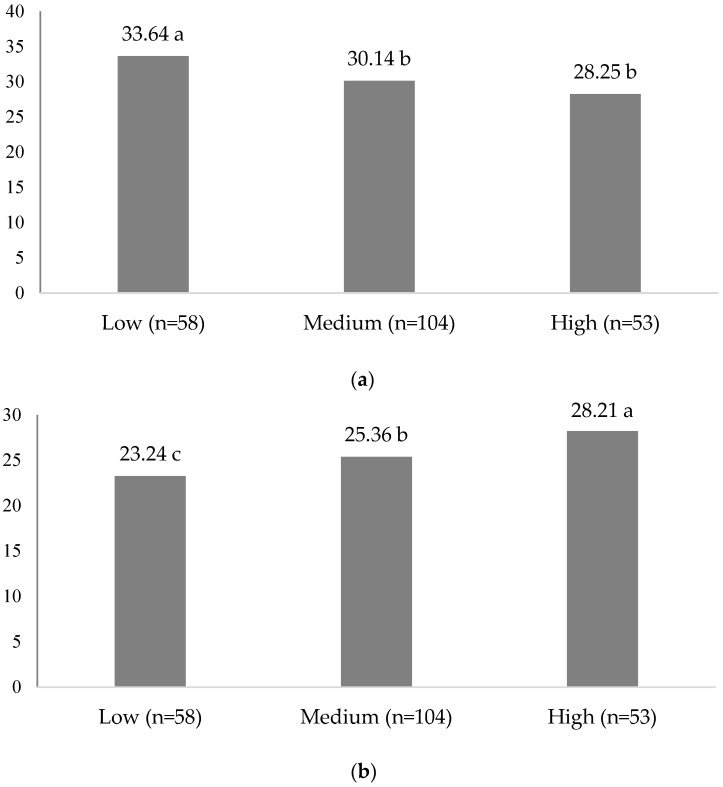
(**a**) FNS and (**b**) VARSEEK scores according to FIS status. Low represents the participants with mean fashion involvement score less than 2.6; medium represents the participants who had mean fashion involvement scores between 2.7 and 4.8; high represents the participants who had mean fashion involvement scores over 4.9. Each category represents 25%, 25–75%, and 75% (Q1, Q2, and Q3) quartile values. Alphabetical letters followed by numbers in (**a**,**b**), represent significant differences at *p* < 0.05.

**Table 1 foods-12-01878-t001:** Demographic information of survey participants (*n* = 215).

Demographics	(%)	Demographics	(%)
		Monthly Income	
Male	49.3	<USD 1000	22.3
Female	50.7	USD 1000–USD 2000	18.6
Age		USD 2000–USD 3000	23.3
<20	15.8	USD 3000–USD 4000	14.9
20–29	16.7	USD 4000–USD 5000	9.3
30–39	16.7	>USD 5000	11.6
40–49	17.2	Region of Residency	
50–59	17.2	Seoul/Gyeonggi region	62.8
60+	16.3	Chungcheong region (mid-central)	7.0
Occupation		Jeolla region (southwest)	6.5
Housewives	8.8	Gyeongsang region (southeast)	21.4
Government	3.7	Gangwon region (northeast)	2.3
Services	7.9	Average BMI of all participants: 23.2 ± 3.8	
Manufacturer	2.8		
Self-employed	6.5		
Professionals	9.8		
Employee	30.7		
Student	19.5		

**Table 2 foods-12-01878-t002:** Participants’ consumption characteristics of food and fashion-related items (N = 215).

Attributes	(%)		Attributes	(%)	
Purchase Frequency	Food	Fashion	Monthly Spending budget	Food	Fashion
Less than once a month	11.2	44.2	<USD 100	34.0	42.8
1–2 times a month	25.6	41.4	USD 100–USD 200	37.7	37.2
3–4 times a month	31.6	9.8	USD 200–USD 300	17.2	12.1
More than 5 times a month	31.6	4.7	USD 300–USD 500	7.9	5.1
Time of purchase	Food	Fashion	USD 500–USD 700	1.9	1.4
Breakfast/Morning (8 am–12 pm)	0.9	3.3	USD 700–USD 1000	1.4	1.4
Lunch/Afternoon (12 pm–4 pm)	20.5	31.6	Factors affecting purchase	Food	Fashion
Dinner/Evening (4 pm–8 pm)	73.0	48.8	Flavor/Design	71.2	49.8
Late night (8 pm–12 am)	4.7	14.9	Price	18.1	27.0
Others	0.9	1.4	Name value (brand)	2.3	5.6
Place of purchase	Food	Fashion	Health/Size	2.3	3.7
Near house/school/workplace	55.8	10.2	Convenience	3.3	12.1
Near Downtown	11.2	13.5	Recommendation	0.5	0.5
Near department stores	0.9	16.3	Trend	0.5	1.4
Online (Mobile apps)	31.2	58.6			
Others	0.9	1.4			

**Table 3 foods-12-01878-t003:** Mean ± standard deviation (SD) of food neophobia and VARSEEK scores.

Food-Related Questionnaires	Mean ± SD
Food Neophobia Scales ^(1)^	
I constantly sample new and different foods ^R^.	2.42 ±1.51
I do not trust new foods.	3.41 ± 1.35
If I do not know what is in a food, I won’t try it.	4.12 ± 1.66
I like foods from different countries ^R^.	2.55 ± 1.55
Ethnic food looks too weird to eat.	3.55 ± 1.45
At dinner parties, I will try a new food ^R^.	2.72 ± 1.60
I am afraid to eat things I have never had before.	3.71 ± 1.69
I am very particular about the foods I will eat.	3.40 ± 1.75
I will eat almost anything ^R^.	2.28 ± 1.69
I like to try new ethnic restaurants ^R^.	2.44 ± 1.45
SUM of neophobia scores	30.61 ± 10.71

^(1)^ Neophobia scale was rated on a 7-point scale, with 1 = completely disagree, 4 = neither disagree nor agree, and 7 = completely agree; ^R^ represents the reverse scale. Therefore, the numbers in this questionnaire were converted into the reverse scale.

**Table 4 foods-12-01878-t004:** Mean ± standard deviation (SD) of VARSEEK scores.

Food-Related Questionnaires	Mean ± SD
VARSEEK Scales ^(1)^	
When I eat out, I like to try the most unusual items, even if I am not sure I would like them.	2.71 ± 0.98
While preparing food or snacks, I like to try out new recipes.	2.99 ± 0.97
I think it is fun to try out food items I am not familiar with.	3.20 ± 1.01
I am eager to know what kind of foods people from other countries eat	3.27 ± 1.07
I like to eat exotic foods.	3.12 ±1.07
Items on the menu that I am unfamiliar with make me curious.	3.17 ± 1.07
I prefer to eat food products I am used to.	3.72 ± 0.93
I am curious about food products I am not familiar with.	3.31 ± 0.98
SUM of VARSEEK scores ^(1)^	25.49 ± 5.37

^(1)^ VARSEEK scale was rated on a 5-point scale, with 1 = completely disagree, 3 = neither disagree nor agree, and 5 = completely agree.

**Table 5 foods-12-01878-t005:** Mean ± standard deviation (SD) of fashion involvement status scores.

Fashion-Related Questionnaires ^(1)^	Mean ± SD
Fashion product involvement	
Fashion clothing means a lot to me.	3.73 ± 1.70
Fashion clothing is a significant part of my life.	3.68 ± 1.69
I am very interested in fashion clothing.	3.90 ± 1.75
I consider fashion clothing to be a central part of my life.	3.93 ± 1.67
I think about fashion clothing a lot.	3.78 ± 1.67
Average of fashion product involvement scores	3.81 ± 1.60

^(1)^ All fashion-related questionnaires were rated on a 7-point scale, with 1 = disagree strongly and 7 = agree strongly.

**Table 6 foods-12-01878-t006:** Correlation between FNS, VARSEEK, and FIS.

Variables	FNS	VARSEEK	FIS
FNS	1	−0.735 *	−0.178 **
VARSEEK		1	0.353 *
FIS			1

* Symbol represents significant differences at *p* < 0.0001; ** symbol represents significant differences at *p* = 0.009.

## Data Availability

Data are only available upon request due to restrictions on privacy or ethical constraints.

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
