# Peer review of "Consumers’ Neophobic and Variety-Seeking Tendency in Food Choices According to Their Fashion Involvement Status: An Exploratory Study of Korean Consumers"

_foods, 2023, doi:10.3390/foods12091878_

Round 1
Reviewer 1 Report
REVIEWER COMMENTS
1. The aim of the study is insufficient. Results could have been given more clearly in the abstract.
2. The abstract should be revised and rewritten in a more understandable way.
3. The purpose should be clearly written in the introduction.
4. Discussions are inadequate and not clear enough. Evaluation should be done in conjunction with current studies.
5. The results in the tables are not clear enough.

Extensive editing of English language required
Reviewer 2 Report
The manuscript is interesting and novel, however, it is necessary to address the following comments:
Line 21: the positive correlation found between VARSEEK and FIS, is it high or low, is it a good value?
Line 28: correct the left indent format in all paragraphs where required, review the Microsoft word template of the authors guide
Line 40: It is necessary to correct from page 2, the margins of the paragraphs according to what is indicated in the Microsoft word template
Line 43: use [4–6] instead [4,5,6]
Line 53: use [12–15] instead [12,13,14,15]
Line 73: the objective of this study was to….
Line 100: Do you consider the total number of volunteers adequate for this study? Is it enough or is a larger number required?
Lines 118,126-137,145,151,165,165,173,179: In the results section, the information should not be discussed in comparison with other studies, this is done in the discussion section. Therefore, it is preferable to determine if the results and discussion section will be presented together or separately. In this context, it is necessary to combine both section for this study
Line 179: use the correct text format to cite references
Line 217: use [4,27,28] instead [4, 27, 28]
Line 251: journal names in each reference should appear in italic text format. Correct through this section
Line 251: the volume should appear in italic text format, not include the issue number...for example...2002, 56, 32–57. It is necessary to correct trough this section
Line 251: use [32–57] instead [32–57]. Correct through this section
Line 255: delete space… K-L.K.;
Line 256: delete the comma…. Manag. 2017, …. Correct format through this section
Line 261: delete space… D.W.
Line 266: correct reference format…. 2011, 29,
Line 272: correct reference format….Korean J. Food Cult. 2011, 26, 429–436.
NOTE: it is necessary to make a thorough review of the appropriate format to cite the references, this can be reviewed in the Microsoft word template
Reviewer 3 Report
In my opinion, the paper is interesting, even if not original at all. The analysis is linked at a specific country with particular features compared to those of many other countries. Yet it can be a good reference point for similar studies to be conducted in other countries. A more adequate introduction and literature review could justify the research. Introduction presents properly the aim of the study, yet the research questions to be addressed are not clearly exposed and, above all, justified by the literature. As a matter of fact, the author/s must include accurate and recent references to support the hypotheses and the study. So, strongly I suggest to consider a more recent and innovative papers on the topic and important in the international context. Research design and methodology could be appropriate, yet different analyses have been conducted which enrich the empirical analysis (so again, the authors must consider further literature): I recommend the author/s to better specify the goodness of the specific quantitative method to support the conceptual model,, I strongly suggest to consider in the text Doi 10.1016/j.foodres.2019.03.027 and its literature. And moreover, why is the used methodology better than other important ones? And besides, are the author/s sure that the sample is representative of the population? Especially interesting is the analyses conducted, but I can say also the results could be more appropriate and clear; moreover, discussion section is relevant and conclusions must resume properly the topic address and the implications for several players. So, really what does the paper add to previous researches? The quality of communication is good and clear enough.
Moderate revision is necessary.
Round 2
Reviewer 3 Report
Now I am satisfied.
Nothing.